# Pannexin-1 Modulates Inhibitory Transmission and Hippocampal Synaptic Plasticity

**DOI:** 10.3390/biom13060887

**Published:** 2023-05-25

**Authors:** Francisca García-Rojas, Carolina Flores-Muñoz, Odra Santander, Pamela Solis, Agustín D. Martínez, Álvaro O. Ardiles, Marco Fuenzalida

**Affiliations:** 1Centro de Neurobiología y Fisiopatología Integrativa, CENFI, Instituto de Fisiología, Universidad de Valparaíso, Valparaíso 2340000, Chile; fjgr14@gmail.com (F.G.-R.); carolina.flores.munoz@gmail.com (C.F.-M.); odra.santander@gmail.com (O.S.); pamela.solisc@postgrado.uv.cl (P.S.); 2Programa de Doctorado en Ciencias, Mención Neurociencia, Universidad de Valparaíso, Valparaíso 2340000, Chile; 3Centro de Neurología Traslacional, Facultad de Medicina, Universidad de Valparaíso, Valparaíso 2341386, Chile; 4Programa de Magister en Ciencias, Mención Neurociencia, Universidad de Valparaíso, Valparaíso 2340000, Chile; 5Centro Interdisciplinario de Neurociencia de Valparaíso, Facultad de Ciencias, Universidad de Valparaíso, Valparaíso 2360102, Chile; 6Centro Interdisciplinario de Estudios en Salud, Escuela de Medicina, Facultad de Medicina, Universidad de Valparaíso, Viña del Mar 2572007, Chile

**Keywords:** pannexin-1, endocannabinoids, GABA transmission, excitatory/inhibitory balance

## Abstract

Pannexin-1 (Panx1) hemichannel is a non-selective transmembrane channel that may play important roles in intercellular signaling by allowing the permeation of ions and metabolites, such as ATP. Although recent evidence shows that the Panx1 hemichannel is involved in controlling excitatory synaptic transmission, the role of Panx1 in inhibitory transmission remains unknown. Here, we studied the contribution of Panx1 to the GABAergic synaptic efficacy onto CA1 pyramidal neurons (PyNs) by using patch–clamp recordings and pharmacological approaches in wild-type and Panx1 knock-out (Panx1-KO) mice. We reported that blockage of the Panx1 hemichannel with the mimetic peptide ^10^Panx1 increases the synaptic level of endocannabinoids (eCB) and the activation of cannabinoid receptors type 1 (CB1Rs), which results in a decrease in hippocampal GABAergic efficacy, shifting excitation/inhibition (E/I) balance toward excitation and facilitating the induction of long-term potentiation. Our finding provides important insight unveiling that Panx1 can strongly influence the overall neuronal excitability and play a key role in shaping synaptic changes affecting the amplitude and direction of plasticity, as well as learning and memory processes.

## 1. Introduction

Pannexin-1 (Panx1) is an integral membrane protein broadly distributed in the brain that forms non-selective ion/metabolite channels, also called hemichannels because they can form intercellular channels similar to gap junction channels [1,2,3]. Emerging evidence suggests that Panx1 hemichannels can contribute to diverse biological processes, including, but not limited to, the development of the central nervous system, synaptic plasticity, learning, and memory [4,5]. In the hippocampus, high Panx1 mRNA and protein levels have been detected in the pyramidal and granular cell layers, consistent of expression in the excitatory neurons [2,6,7,8], but also in GABAergic interneurons within the stratum oriens, radiatum, and lacunosum-moleculare [2,7,8]. Notably, Panx1-KO mice exhibit enhanced long-term potentiation (LTP) and excitability in the hippocampus, suggesting that Panx1 hemichannels regulate the strength of glutamatergic synaptic plasticity [4,5,9]. Panx1 loss-of-function affects LTP and LTD by altering the contribution of GluN2 receptors to NMDAR-dependent synaptic plasticity in the hippocampus [10]. Consistently, Panx1-KO mice present memory and learning alterations, manifested as impaired object recognition and spatial memory [5,10]. Interestingly, a recent study suggests that metabotropic NMDAR signaling through Src kinase regulates postsynaptic Panx1 hemichannels in hippocampal PyNs [11]. Panx1 hemichannels also participate in the post-synaptic buffering of anandamide (AEA), since Panx1 hemichannel blockade and deletion increase the extracellular levels of AEA and reduce TRPV1-dependent glutamate release [11].

Though this evidence clearly shows the importance of the functional interaction between Panx1 hemichannels and glutamatergic synaptic transmission, it remains undetermined whether Panx1 hemichannels could regulate GABAergic synaptic plasticity. GABAergic interneurons exert strong and precise control over the activity of glutamatergic neurons, which is critical to maintaining the excitatory–inhibitory (E/I) balance, providing the proper circuit activity for cognitive function [12,13,14,15]. Although is clear that GABA synapses exhibit several forms of plasticity [16,17,18], how inhibitory synaptic plasticity is induced and expressed at the diverse synapses on the principal neurons is only beginning to be unraveled [19,20]. Here, we report that the Panx1 hemichannel is a relevant modulator of GABAergic synaptic transmission. Blocking Panx1 hemichannels led to a decrease in GABAergic synaptic efficacy. The phenomenon involves the eCBs system, postsynaptic free Ca^2+^, and the PKA signaling pathway. The present research highlights the key role of Panx1 hemichannels as part of a pathway that controls GABAergic synaptic transmission, E/I balance, and threshold for LTP, all of which are fundamental for regulating hippocampal plasticity, learning, and memory.

## 2. Materials and Methods

### 2.1. Animals

Animal care procedures and slice preparation met the National Institute of Health (US) guidelines and were approved by the Institutional Animal Ethics Committee of Universidad de Valparaíso (document number: BEA160-20). Male and female adult C57BL/6J wild-type (WT) and Panx1-knockout (Panx1-KO) mice [4,21] were housed on a 12 h light/dark cycle in a temperature- and humidity-controlled environment with ad libitum access to food and water.

### 2.2. Chemicals and Reagents

All reagents were obtained from Sigma-Aldrich Chemistry (St. Louis, MO, USA), Tocris Bioscience (Bristol, UK), Merck (Newark, NJ, USA), and Cayman Chemical (Ann Arbor, MI, USA). Selective Panx1 inhibitor ^10^Panx1 (WRQAAFVDSY) and its scrambled control peptide sc^10^Panx1 (FSVYWAQADR) were synthesized by Beijing SBS Genetech Co., Ltd. (Beijing, China). The C-terminal polyclonal antibody anti-Panx1 (α-panx1; 0.25 ng/µL) [11,22,23] was employed to block postsynaptic Panx1 hemichannels and was purchased from Invitrogen (rabbit polyclonal, #488100, Whaltham, MA, USA). Stock solutions were prepared in miliQ water (2-amino-5-phosphonopentanoid acid; APV, 7-nitro-2,3-dioxo-1,4-dihydroquinoxaline-6-carbonitrile; CNQX and ^10^Panx1/sc^10^Panx1) or DMSO (Dihydrochloride; H89, AM251, WIN 55,212,2; WIN, capsaicin; CAP, and capsazepine; CPZ) and added to the ACSF as needed.

### 2.3. Slice Preparation

Wild-type and Panx1-KO mice were anesthetized with isoflurane, decapitated, and their brain was rapidly removed through craniotomy and submerged in cold (~1 °C) artificial cerebrospinal fluid (ACSF, in mM: 124.00 NaCl, 2.70 KCl, 1.25 KH_2_PO4, 1.30 MgSO_4_, 26.00 NaHCO_3_, 10.00 Glucose, 2.50 CaCl_2_). The pH of the ACSF was stabilized at 7.4 by bubbling with a mixture of oxygen (95%) and CO_2_ (5%). Coronal slices from the dorsal hippocampus (300 µM) were cut with a vibratome (Campden Instruments, model MA752, (Loughborough, UK), incubated in ACSF (>1 h at room temperature; 20–22 °C), and then transferred to a 1 mL chamber fixed to a magnifying glass. Slices were continuously perfused with ACSF and maintained at room temperature (20–22 °C).

### 2.4. Electrophysiology

Whole-cell recordings from the soma of CA1 pyramidal neurons (PyNs) were obtained by using borosilicate glass microelectrodes (4–6 MΩ) filled with an internal solution containing (in mM) 131 Cs-gluconate, 10 HEPES, 10 EGTA, 2 glucose, 1 CaCl_2_, 8 NaCl, 4 Mg_2_ATP, and 0.4 Na_3_GTP buffered to pH 7.2–7.3 with CsOH. For some experiments, the intracellular solution included the fast calcium ion chelator 1,2-bis (o-amino phenoxy) ethane-*N*,*N*,*N*′,*N*′-tetraacetic acid (BAPTA, 40 mM), the membrane-impermeable PKA inhibitor PKI6-22 peptide (2.5 mM), or the C-terminal anti-Panx1 polyclonal antibody (α-panx1; 0.25 ng/µL). In some experiments, the intracellular solution contained the G-protein coupled receptor (GPCR) inhibitor guanosine 5′-*O*-(2-thiodiphosphate) (GDPβS, 2 mM contained (in mM): 110 CsCl, 20 TEA-Cl H_2_O, 10 HEPES, 1 MgCl_2_6H_2_O, 5 EGTA, 4 2Na-ATP. The holding potential (Vh) was adjusted to 0 mV to record inhibitory postsynaptic currents (IPSCs), and the capacitive currents were compensated to 70%. Neurons were only considered when the seal resistance was >1 GΩ, and the access resistance (Ra; 7–14 MΩ) did not change by >20% during the experiment. Data were low-passed, filtered at 3.0 kHz, sampled at 6.0–10.0 kHz using an A/D converter (ITC-16, InstruTech, MA, USA), and acquired with the Pulse Fit software (Heka Instrument, Westfield, MA, USA). An EPC-7 patch–clamp amplifier was used for all recordings (Heka Instrument, Westfield, MA, USA). An epoxy electrode (World Precision Instruments, Sarasota, FL, USA) filled with KCl and a 2.7 KΩ resistance was employed as a reference.

Experiments started after a 5–10 min stabilization period following whole-cell access. Then, evoked and spontaneous IPSCs (eIPSCs and sIPSCs, respectively) were recorded. eIPSCs were elicited using a concentric electrode (platinum/iridium, FHC Inc., Bowdoin, ME, USA) placed at the pyramidal layer (~10–20 μm). GABAergic terminals were activated by bipolar cathodic stimulation through an isolation unit (Isoflex, AMPI, Jerusalem, Israel). Recordings of miniature inhibitory postsynaptic currents (mIPSC) were performed under tetrodotoxin (TTX) (1 μM). Pharmacological reagents were bath applied after the establishment of a stable baseline (~10–15 min), and potential effects recorded after responses were stable (>15 min). eIPSC amplitude was calculated as a percentage of change, considering 5 min of baseline and 5 min after the drug perfusion. Voltage–clamp data were high-pass filtered at 3.0 kHz, sampled at 6.0–10.0 kHz through an A/D converter ITC-16 and acquired using Pulse Fit (Heka Instrument, Westfield, MA, USA). Paired-pulse ratio (PPR) was calculated as (R2/R1) × 100, where R1 and R2 were the peak amplitudes of the first and second IPSCs, respectively.

Recordings of field EPSPs (fEPSPs) were performed with a glass pipette (2–4 MΩ, filled with ACSF) placed in the middle of the stratum radiatum in CA1. The recording electrode was connected to an AC amplifier (P-5 series; Grass) with a gain of 10,000, a low-pass filter of 3.0 kHz, and a high-pass filter of 0.30 Hz. Schaffer collateral fibers were activated by bipolar cathodic stimulation generated by a Master 8 stimulator (AMPI) and connected to an isolation unit (Isoflex; AMPI, Jerusalem, Israel). The bipolar concentric electrode (platinum/iridium; FHC, Bowdoin, ME, USA) was placed in the stratum radiatum 100–200 µm away from the recording site. Electric pulses (50 µs, 0.3 Hz, and 20–100 µA) were applied to the Schaffer collaterals. Basal excitatory synaptic transmission was measured using an input/output curve protocol [24], consisting of eight stimuli, ranging from 200 to 900 µA (interval between stimuli 10 s). A high-frequency stimulation (HFS) protocol, consisting of 1 train of 100 pulses at 100 Hz for 1 s and separated by 1 s applied after at least 15 min of stable baseline recordings, was employed to elicit LTP. All experiments were conducted at room temperature (20–22 °C).

### 2.5. Analysis

Data analysis was performed using the software Clampfit (10.7.0.3) from pCLAMP 10.7 (Molecular Devices, San José, CA, USA). GraphPad Prism 8.0.1 for Windows was used for statistical analysis and graphic construction. We verified normal distribution using Shapiro Wilk (SW). Data with a normal distribution were analyzed with a two-tailed *t*-test. Wilcoxon matched-pairs signed rank test was used for paired, nonparametric data. Unless otherwise stated, all data are expressed as mean ± SEM. Differences were considered statistically significant at *p* < 0.05.

## 3. Results

### 3.1. The Blockage of Panx1 Hemichannel Decrease Hippocampal GABAergic Efficacy

To explore the potential contribution of Panx1 to GABAergic inhibitory synapses, we selectively blocked Panx1 hemichannels with the mimetic peptide ^10^Panx1 (100 µM), and we determined its effect on spontaneous and evoked GABAergic synaptic transmission in PyNs. ^10^Panx1 reduced eIPSC amplitude (baseline: 100.22 ± 1.55%, 10Panx: 73.33 ± 6.86%; *p* = 0.0055, *n* = 10 per condition; Figure 1A) and increased paired-pulse ratio (PPR; control: 0.55 ± 0.03, *n* = 10; ^10^Panx1: 0.66 ± 0.05; *p* = 0.0048, *n* = 10; Figure 1A, right), suggesting that Panx1 hemichannel blockade causes a reduction in GABA release onto hippocampal CA1 PyNs. These effects were not observed upon the application of a scrambled version of the peptide, sc^10^Panx1 (100 µM) (for IPSC amplitude, baseline: 98.90 ± 1.98%, sc10Panx: 93.90 ± 10.47%; *p* = 0,6613, *n* = 7 per condition; for PPR, baseline: 0.59 ± 0.03, sc^10^Panx1: 0.62 ± 0.04; *p* = 0.2421, *n* = 9 per condition; Figure 1B), confirming the specificity of ^10^Panx1.

To determine if the reduction in GABAergic transmission observed after inhibiting Panx1 hemichannels is postsynaptic, we selectively inhibit Panx1 hemichannels in single CA1 PyNs by adding α-Panx1 0.25 ng/µL (anti-Panx1 antibody) in the recording pipette [11,22,23]. Incubation with ^10^Panx1 did not change eIPSC amplitude in PyNs with intracellular blockage of Panx1 hemichannels (ACSF with α-panx1: 100.79 ± 1.39%, α-panx1 + 10Panx: 93.67 ± 7.53%; *p* = 0.4543, *n* = 6 per condition; Figure 1C), nor PPR (ACSF with α-panx1: 0.72 ± 0.052, α-panx1 + 10Panx: 0.70 ± 0.050; *p* = 0.4929, *n* = 6 per condition; Figure 1C, right). As a control for intracellular α-panx1, we added a polyclonal antibody against α-Cx43 (0.3 ng/μL) in the recording pipette solution. We found that PPR values in the presence of α-Cx43 were significantly lower compared to the α-panx1 condition (α-Cx43: 0.40 ± 0.06; *n* = 6, α-panx1: 0.62 ± 0.04; *p* = 0.0140, *n* = 10; Appendix A). This finding suggests that postsynaptic Panx1 hemichannel blockade with α-panx1 is sufficient to decrease basal GABAergic efficacy.

To further confirm the Panx1 hemichannel contribution to GABAergic transmission and to test the specificity of ^10^Panx1, we assessed the effects of its application in hippocampal slices from Panx1-KO mice. As expected, ^10^Panx1 affected neither the IPSC amplitude (ACSF: 99.82 ± 1.25%, ^10^Panx1: 99.34 ± 10.93%; *p* = 0.9825, *n* = 5 per condition; Figure 1D, left), nor PPR (ACSF: 0.61 ± 0.05, 10Panx: 0.58 ± 0.06; *p* = 0.5320, *n* = 5 per condition; Figure 1D, right) in PyNs from Panx1-KO mice, suggesting that ^10^Panx1 effects are mediated by Panx1 hemichannel blockade and not by other mechanisms. During the experiment, no changes in the access resistance were observed (Figure 1A–D).

Additionally, we measured the inhibitory efficacy of CA1 PyNs from Panx1-KO to evaluate whether the absence of Panx1 alters basal GABAergic transmission. The lack of Panx1 did not lead to changes in the amplitude of evoked IPSCs (WT: 413.702 ± 55.06 pA; *n* = 15, Panx1-KO: 319.75 ± 44.13 pA; *p* = 0.2329, *n* = 10; Appendix A) or the PPR (WT: 0.3805 ± 0.03; *n* = 15, Panx1-KO: 0.453 ± 0.06; *p* = 0.2918, *n* = 10; Appendix A). Basal inhibitory transmission (spontaneous and miniature events) was comparable among pyramidal cells from Panx1-KO and WT mice (Appendix A).

The presynaptic origin of ^10^Panx1-mediated depression in GABAergic efficacy was confirmed by analyzing the effect of the Panx1 hemichannel blockade on spontaneous IPSCs (sIPSCs). We observed that the application of ^10^Panx1 decreased sIPSC frequency (from 3.86 ± 0.33 Hz to 2.83 ± 0.32 Hz; *p* = 0.0001, *n* = 8; Figure 2A,B), while sIPSC amplitude was unaffected (changed from: 38.28 ± 4.72 pA, to 32.65 ± 3.23 pA; *p* = 0.1511, *n* = 8; Figure 2A). Several reports suggest that Panx1 hemichannels may regulate hippocampal excitability [4,5,9,11], which could be related to the decrease in sIPSC frequency. To assess the involvement of Panx1 hemichannels blockade on action potential-dependent synaptic events, we inhibited the generation of action potential with TTX (500 nM) and recorded miniature IPSC (mIPSC) for 10–15 min after the ^10^Panx1 application. The mimetic Panx1 peptide reduced the frequency of mIPSC (from: 1.97 ± 0.33 Hz to 1.36 ± 0.27 Hz; *p* = 0.0010, *n* = 8 per condition; Figure 2B), without affecting the amplitude of mIPSC (from: 15.36 ± 0.73 pA, to 14.96 ± 1.18 pA; *p* = 0.5136, *n* = 8 per condition; Figure 2B). In addition, we observed that cells loaded with α-Panx1 showed a significant decrease in the frequency of sIPSCs compared with their control, α-Cx43 (α-Cx43: 4.48 ± 0.05 Hz, α-Panx1: 2.58 ± 0.43 Hz; *p* = 0.0197, *n* = 6 per condition; Figure 2C), while sIPSCs amplitude remained unchanged (α-Cx43: 28.42 ± 3.57 pA, α-Panx1: 24.22 ± 3.74 pA; *p* = 0.4368, *n* = 6 per condition; Figure 2C).

### 3.2. ^10^Panx1-Induced Depression of Inhibitory Transmission Requires CB1R and TRPV1 Activation and Presynaptic cAMP/PKA

Different types of GABAergic functional and structural plasticity have been reported at inhibitory synapses on principal cells [25]. The best-characterized form of presynaptic inhibitory plasticity in the hippocampus and other brain regions is the eCB-mediated I-LTD [26,27,28]. Panx1 hemichannels are reversibly activated by multiple Gαq-coupled receptors and can increase Ca^2+^ influx through interactions with NMDAR, P2X, and TRPV4 [29]. Anandamide, an endogenous cannabinoid, can activate both CB1R and TRP channels, bidirectionally modulating synaptic transmission. It has been proposed that Panx1 hemichannels can regulate the synaptic concentration of eCB and affect synaptic transmission [11]. Moreover, a recent report shows that activation of metabotropic NMDAR pathways facilitates glutamate release in hippocampal synapses through AEA accumulation and activation of presynaptic TRPV1 [11]. To determine whether Panx1 hemichannels blockade decreases GABA transmission, involving the eCBs signaling, first, we inhibited CB1R and then applied ^10^Panx1. In the presence of the selective CB1 receptor antagonist AM251 (5 µM), which has been shown not to impact GABA transmission (Appendix A), the amplitude of evoked IPSCs (eIPSCs) was not significantly altered by the application of 10Panx1 (AM251: 100.70 ± 1.22%, AM251 + ^10^Panx1: 103.44 ± 13.41%; *p* = 0.8503, *n* = 6 per condition; Figure 3A). Additionally, ^10^Panx1 did not affect the PPR (AM251: 0.536 ± 0.07, AM251 + ^10^Panx1: 0.517 ± 0.07; *p* = 0.2641, *n* = 7 per condition; Figure 3A). If Panx1 hemichannels blockade increases the synaptic levels of eCBs and decreases the GABA transmission, we reasoned that the exogenous cannabimimetic WIN 55,212-2 (WIN) that depress the GABA transmission (Appendix A) should occlude this form of plasticity. In slices pretreated with WIN (5 µM), the GABAergic depression induced by ^10^Panx1 was completely occluded. ^10^Panx1 did not affect eIPSC amplitude (WIN: 99.28 ± 1.787%, WIN + ^10^Panx1: 103.98 ± 13.76%, *p* = 0.7422, *n* = 6; Figure 3B), nor the PPR (WIN: 0.722 ± 0.08; WIN + ^10^Panx1: 0.692 ± 0.04; *p* = 0.6208, *n* = 6 for each condition; Figure 3B). The present results show that the GABAergic depression induced by ^10^Panx1 requires CB1R activation.

Since AEA is also an agonist of the TRPV1 [30] and Panx1 hemichannels permeate AEA [11], we examined whether TRPV1 could be involved in the depression of GABAergic transmission mediated by Panx1 hemichannel blockade. In the presence of the specific TRPV1 antagonist capsazepine (CPZ, 10 µM), bath application of ^10^Panx1 did not affect IPSC amplitude (CPZ: 98.28 ± 0.88%, CPZ + ^10^Panx1: 106.62 ± 7.29%; *p* =0.3007, *n* = 7 per condition; Figure 3C), nor PPR (CPZ: 0.58 ± 0.06, CPZ + ^10^Panx1: 0.57 ± 0.04; *p* = 0.7461, *n* = 7; Figure 3C). Consistent with these results, pharmacological activation of TRPV1 with the specific agonist capsaicin (CAP, 1 µM) also occluded the depression of GABAergic transmission induced by ^10^Panx1, maintaining eIPSC amplitude (CAP: 98.82 ± 1.00%, CAP + ^10^Panx1: 114.48 ± 9.42%; *p* = 0.1321, *n* = 6 per condition; Figure 3D), and PPR (CAP: 0.57 ± 0,04, CAP + ^10^Panx1: 0.57 ± 0.03; *p* = 0.8189, *n* = 6; Figure 3D) was unchanged after 10Panx application. These findings indicate that activation of the TRPV1 is also required for the GABAergic transmission depression induced by ^10^Panx1.

eCB-dependent long-term depression in the hippocampus requires the downregulation of AC/cAMP/PKA signaling and the active-zone scaffolding protein RIM1α [27,31], which is a PKA target protein [32]. Thus, we analyzed whether presynaptic AC/cAMP/PKA signaling was involved in the induction of inhibitory depression caused by ^10^Panx1. First, we tested the participation of PKA using the cell permeable PKA inhibitor H89 (10 µM). We showed that 20 min of preincubation with H89 was enough to occlude the effect of ^10^Panx1 on eIPSC amplitude (H89: 101.27 ± 2.15%; H89 + ^10^Panx1: 75.90 ± 10.96%, *p* = 0.1069, *n* = 5 per condition, Figure 4A) and PPR (PPR in H89: 0.68 ± 0.03, H89 + 10Panx: 0.690 ± 0.04; *p* = 0.6024, Figure 4A). Since H89 is a cell-permeant PKA inhibitor, it could affect both pre-and post-synaptic signaling. To specifically study the postsynaptic involvement of PKA in ^10^Panx1-mediated GABAergic depression, we added the non-myristoylated membrane impermeant form of the PKA inhibitor-peptide PKI_6-22_ (2.5 μM) to the recording pipette. PKI_6-22_ completely inhibited the effect of ^10^Panx1 on PPR (ACSF: 0.47 ± 0.04, ^10^Panx1: 0.56 ± 0.05; *p* = 0.049; *n* = 6 per condition, Figure 4B). Moreover, the effect of ^10^Panx1 in eIPSCs amplitude was prevented by the presence of PKI_6-22_ (ACSF: 98.76 ± 2.47%, ^10^Panx1: 102.16 ± 5.16%; *p* = 0.499, *n* = 6 per condition, Figure 4B). These findings indicate that ^10^Panx1 reduced GABAergic transmission through the downregulation of both the pre-and post-synaptic AC/cAMP/PKA pathway.

### 3.3. ^10^Panx1-Induced Inhibitory Depression Requires a Rise in Postsynaptic Intracellular Ca^2+^

eCBs-mediated changes in synaptic efficacy can be generated by two different cellular mechanisms: (a) a strong increase in postsynaptic free Ca^2+^ levels and (b) activation of postsynaptic Gq11-protein-coupled receptors [33,34]. First, we determined if the depression of GABAergic transmission caused by ^10^Panx1 requires changes in postsynaptic free Ca^2+^ levels. The addition of the Ca^2+^ chelator BAPTA (20 mM) to the patch pipette prevented the effects of ^10^Panx1 on eIPSCs amplitude (ACSF: 99.46 ± 1.22%, ^10^Panx1: 105.09 ± 10.25%; *p* = 0.5777; *n* = 7 per condition, Figure 5A) and PPR (ACSF: 0.60 ± 0.02, ^10^Panx1: 0.68 ± 0.05; *p* = 0.1281, *n* = 7 per condition, Figure 5A).

l-type voltage-gated calcium channels play a major role in depolarization-induced Ca^2+^ influx in hippocampal PyNs [35,36]. To determine their potential involvement in ^10^Panx1-induced GABAergic depression, we blocked l-type Ca^2+^ VGCC with nimodipine (NMD, 10 µM). In the presence of NMD, the addition of ^10^Panx1 did not modify eIPSCs (NMD: 98.67 ± 1.07%, NMD + 10Panx: 98.17 ± 11.78%; *p* = 0.9695; *n* = 6 per condition, Figure 5B), nor the PPR (NMD: 0.64 ± 0.06, NMD + 10Panx: 0.62 ± 0.05; *p* = 0.3203; *n* = 6 for each condition, Figure 5B), suggesting that l-type Ca^2+^ VGCC are involved in the depression of the GABAergic transmission caused by ^10^Panx1. Ischemia and prolonged NMDA exposure activate Panx1 hemichannels [22]. This downstream Panx1 hemichannel activation is mediated by sarcoma (Src) kinases and requires metabotropic-mediated NMDAR activity [23]. Due to the functional interaction between NMDAR and Panx1 hemichannel, we studied whether ionotropic glutamate receptors (i.e., AMPAR/NMDARs) play a role in the depression of GABAergic transmission caused by ^10^Panx1. In the presence of APV and CNQX, ^10^Panx1 reduced eIPSC amplitude (APV/CNQX: 99.52 ± 0.99, APV/CNQX+10Panx: 76.37 ± 4.13%; *p* = 0.0024; *n* = 7 per condition, Figure 5C) and increased PPR (APV/CNQX: 0.50 ± 0.03, APV/CNQX+^10^Panx1: 0.62 ± 0.02; *p* = 0.0134, *n* = 7 for each condition, Figure 5C), suggesting that ^10^Panx1-mediated GABAergic depression is NMDAR and AMPAR independent. In addition, we tested if postsynaptic G-protein activation at the PyNs was required for the induction of ^10^Panx1-mediated GABAergic depression. Adding the G-protein inhibitor GDPβS (2 mM) to the recording pipette failed to block the GABAergic depression induced by ^10^Panx1. Under GDPβS, 10Panx decreases the eIPSCs amplitude (ACSF: 96.29 ± 0.97%, ^10^Panx1: 53.33 ± 6.29%, *p* = 0.0015, *n* = 6 per condition, Figure 5D) and increases the PPR (ACSF: 0.56 ± 0.02, 10Panx: 0.64 ± 0.03, *p* = 0.0082, *n* = 6 for each condition, Figure 5D), ruling out the involvement of G protein-coupled receptors in this form of inhibitory synaptic depression.

### 3.4. Involvement of Panx1 Hemichannels in the Regulation of the E/I Ratio and the Threshold for Synaptic Plasticity in CA1 Pyramidal Neurons

The balance between excitation and inhibition is essential for regulating brain activity [12]. Synaptic plasticity confers a dynamic way to balance excitatory and inhibitory inputs precisely and efficiently at the target cell during learning processes [37,38]. It has been shown that blocking Panx1 hemichannels in single postsynaptic CA1 neurons produces an increase in glutamate release, which modifies network excitability [11]. So far, we have shown that Panx1 hemichannel blockade depresses inhibitory transmission, suggesting that Panx1 hemichannels could play a pivotal role in regulating the E/I balance.

To determine the effect of Panx1 hemichannel blockade on the setting of E/I balance, we tested the effect of ^10^Panx1 on paired, compound EPSC/IPSC elicited by stratum radiatum stimulation and we recorded the CA1 PyNs at −40 mV holding potential in the presence of APV (50 µM; Figure 6). This approach compares the synaptic inputs to the CA1 PyNs by the direct excitatory pathway and the indirect inhibitory pathway via interneurons, which allows calculating the E/I ratio [39]. We found that ^10^Panx1 shifted the E/I ratio. In the control conditions, the E/I ratio was 0.60 ± 0.15, while upon ^10^Panx1 treatment, its value was 1.43 ± 0.52 (*p* = 0.0089, *n* = 7, Figure 6A). The blockade of Panx1 hemichannels shifted the E/I toward excitation, but the inhibitory component was also strongly decreased with ^10^Panx1(Figure 6A), suggesting that Panx1 hemichannels could be essential to maintain an appropriate E/I balance at the hippocampal network.

Previously, we have shown that the blockade of Panx1 hemichannels or absence of Panx1 (Panx1-KO) restrains the sliding threshold for the induction of synaptic plasticity, increasing LTP in the hippocampus [4]. Considering that GABAergic inhibition regulates the excitability and induction of synaptic plasticity [40,41], we hypothesized that ^10^Panx1-mediated depression of GABAergic transmission that alters the E/I balance could reduce the threshold of induction of LTP in CA3-CA1 synapses. First, we compared the effectiveness of single HFS to induce LTP in CA3-CA1 synapses. After 15 min of stable baseline, a single HFS failed to induce LTP (105.49 ± 2.90%; *p* = 0.184; *n* = 6; Figure 6C), whereas two HFSs were able to induce a strong LTP (150.49 ± 13.17%; *p* = 0.010; *n* = 6; Figure 6C). In the presence of ^10^Panx1, a single HFS was sufficient to induce a strong synaptic potentiation (200.17 ± 18.66%; *p* = 0.010; *n* = 6; Figure 6C). Together, our data suggest that Panx1 can regulate inhibitory transmission and excitability, controlling the nature of the postsynaptic signal necessary to induce plasticity in the hippocampus, according to a critical role in brain plasticity and neural network homeostasis and refinement.

## 4. Discussion

This study provides evidence that neuronal Panx1 hemichannels form part of a fundamental pathway that modulates the strength of GABAergic transmission, the E/I balance, and CA1 hippocampal plasticity. Specifically, our result suggests that Panx1 hemichannels blockade with the mimetic peptide ^10^Panx1 can increase the synaptic level of eCBs and the activation of CB1Rs, which results in a decrease in GABAergic efficacy onto CA1 PyNs, shifts the E/I balance toward excitation, and facilitates the induction of long-term potentiation in the hippocampus.

In addition to its significant role in synaptogenesis, Panx1 is enriched in pre-and postsynaptic compartments, regulating glutamatergic synaptic transmission and plasticity [8,9,42]. Previously, it has been demonstrated that Panx1 hemichannel blockade enhances LTP and prevents LTD induction in adult mice [4]. Furthermore, it was found that Panx1-KO mice displayed enhanced excitability, persistent LTP responses, and impaired spatial learning [10]. Additionally, we have demonstrated that Panx1-KO CA1 neurons exhibited enhanced excitability, higher dendritic arborization, and more mature dendritic spines, as well as an increased size of the readily releasable pool of glutamatergic vesicles [9]. Recently, it has been demonstrated that Panx-1 activity drives purinergic signaling-mediated regulation of hyperpolarization-activated cyclic nucleotide (HCN)-gated channels, leading to decreased neuronal excitability and hippocampal network activity [43]. Those results highlight the crucial role of Panx1 hemichannels in paracrine signaling, regulating excitatory synaptic transmission at both the functional and structural levels. Nonetheless, these previous studies did not involve inhibitory synaptic transmission.

GABAergic interneurons modulate excitatory signaling, controlling spike generation and timing [44] and conferring brain circuits with extraordinary flexibility by modulating the gain of pyramidal neuron responses [14,15]. By regulating E/I balance, GABAergic synaptic plasticity can modulate excitability and neural circuit refinement and contribute to the learning and memory process [12,45]. Although Panx1 is also localized to GABAergic interneurons [2,8], its role in GABA synaptic transmission is unknown. Our study showed that blocking Panx1 hemichannels reduced GABA release and eIPCS amplitude, suggesting pre-and postsynaptic changes.

The best-characterized form of presynaptic inhibitory plasticity in the hippocampus and other brain regions is the eCB-mediated I-LTD [26,27,28]. N-arachidonoylethanolamide (anandamide, AEA) and 2-arachidonoylglycerol are two of the most studied eCBs [26,46,47]. Upon demand, typically by activation of certain G protein-coupled receptors or by depolarization, eCBs are liberated and travel backward to activate presynaptic CB1Rs or TRPV1. CB1Rs are Gi/o-coupled receptors whose activation results in the suppression of neurotransmitter release in glutamatergic and GABAergic synapses [26,33,34]. In the adult hippocampus, CB1Rs are mainly localized at synaptic terminals of GABAergic interneurons [48,49]. Our results suggest that the decrease in GABAergic transmission induced by ^10^Panx1 involves eCBs signaling. In support of this possibility, the CB1R blockade with AM251 completely prevented the ^10^Panx1-induced decrease in GABAergic transmission onto PyNs. In addition, we observed that the application of the CB1R agonist WIN occluded the effect of ^10^Panx1, indicating that Panx1 hemichannels act as a regulatory mechanism that buffers eCB accumulation and regulates inhibitory transmission.

It has been demonstrated that the activation of CB1Rs can induce a long-lasting reduction in GABA release by the downregulation of the adenylyl cyclase and cAMP/PKA pathways via the α_i/o_ subunits [31]. We observed that the PKA inhibitor, H89, occluded the ^10^Panx1-mediated depression of the inhibitory transmission. Moreover, we showed that a decrease in the GABA release probability was still observed under the postsynaptic inhibition of PKA with PKI6-22 peptide. These results suggest that the blockage of Panx1 hemichannels boost CB1Rs signaling, which, through the downregulation of presynaptic cAMP/PKA signaling, is necessary for the depression of GABA transmission onto PyNs in mouse hippocampus. The PKA can modulate transmitter release through phosphorylation of presynaptic proteins involved in exocytosis, such as synapsin I, rabphilin-3A, SNAP-25, and α-SNAP [50,51,52], or active zone proteins, such as RIM1α [31].

The synthesis and release of eCBs depend on two main mechanisms: postsynaptic depolarization, increasing the postsynaptic Ca^2+^ via activation of VGCCs, and activation of Gq-protein coupled metabotropic receptors [53,54]. Our findings indicate that blockage of Panx1-dependent eCB release required the activation of postsynaptic l-type VGCCs, suggesting that the postsynaptic free Ca^2+^ increase is essential for the induction of ^10^Panx1-mediated depression of GABAergic transmission. Although we have shown that L-type calcium channels are necessary for this form of inhibitory plasticity, other sources of intracellular calcium may be involved. Recent research has unveiled a noteworthy finding regarding the influence of postsynaptic PKA on the synthesis of 2-AG in striatal medium spiny neurons. This study further reveals that such activity has the potential to inhibit synaptic transmission in the cortico-striatal circuit [54]. Moreover, our results indicate that ^10^Panx1-induced GABAergic depression is G protein-coupled receptors independently, as GDP-βS did not modify the effects of ^10^Panx1 on GABAergic efficacy. Recently, it was demonstrated that NMDAR-Panx1 signaling in the postsynaptic compartment is involved in the homeostatic regulatory mechanism that buffers AEA accumulation from the synaptic space and suppresses the TRPV1-dependent glutamate release [11]. In contrast, our results show that both NMDAR and AMPAR are not involved in the ^10^Panx1-induced depression of inhibitory transmission, suggesting that the postsynaptic free Ca^2+^ required for this depression comes from a different source.

Our findings revealed that the blockade of Panx1 hemichannels not only has a presynaptic effect, but also modifies the efficacy of GABAergic transmission postsynaptically, resulting in a decrease in the amplitude of the eIPSCs. Several forms of inhibitory plasticity occur due to postsynaptic changes in GABAergic transmission, manifested through several mechanisms [55,56]. For instance, increases or decreases in channel function can occur as a result of GABAAR phosphorylation by multiple kinases, including PKA, CaMKII, and Src [57]. It has been reported that there is a differential modulation of GABAAR subtypes by PKA activity, dependent upon β subunit identity. While PKA negatively modulates GABAAR that incorporates β1 subunits, PKA phosphorylation increases the activity of GABAAR, containing the β3 subunit [57,58].

It was recently suggested that Panx1 hemichannel activity induced by mechanical stretch could be reduced by adenosine via a PKA-dependent pathway [59]. Our results showed that, in the presence of postsynaptic PKI6-22, ^10^Panx1 was ineffective in reducing eIPSC amplitude (Figure 4). Those results suggest that postsynaptic PKA is involved in the ^10^Panx1-induced depression of GABAergic efficacy. Yet, interactions between Panx1 and PKA may involve other mechanisms, since GDP-βS did not occlude the effect of ^10^Panx1. In addition, it has been described that inhibitory strength is dependent on the cellular gradient controlled by the function of Cl^−^ transporters. Due to Panx1 hemichannels being anion-selective (~50–80 pS; [60]), which can permeate chloride (Cl^−^), the effect of ^10^Panx1 on GABAergic efficacy could also involve changes in Cl^−^ cellular gradient.

Since we conducted the experiments at room temperature, it is possible that the magnitude and kinetics of the observed effects may differ under physiological temperature conditions. In fact, previous research [61] has shown that temperature increases the release of ATP from cells. Furthermore, it has been suggested that temperature also enhances the formation and function of intercellular channels formed by Panx1 in oligodendrocyte cell lines [3]. The authors observed a consistent temperature-dependent increase in Gap Junctional Conductance (gj) in these cells. Similarly, a previous study [62] demonstrated a temperature-dependent increase in hemichannel currents mediated by endogenous Cx38 in Xenopus oocytes. Therefore, we can speculate that the Panx1-mediated control of GABAergic transmission observed in acute experiments at room temperature may underestimate its actual magnitude under physiological temperatures.

## 5. Conclusions

GABAergic synaptic inhibition finely regulates the balance between excitatory and inhibitory neuronal activity, which plays an important role in neuronal network activity and synaptic plasticity. Here, we showed that Panx1 hemichannel blockade shifts the E/I balance toward excitation and reduces the threshold for LTP induction, suggesting that Panx1 hemichannels are an important element of a pathway that regulates of synaptic strength in the adult hippocampus.

## Figures and Tables

**Figure 1 biomolecules-13-00887-f001:**
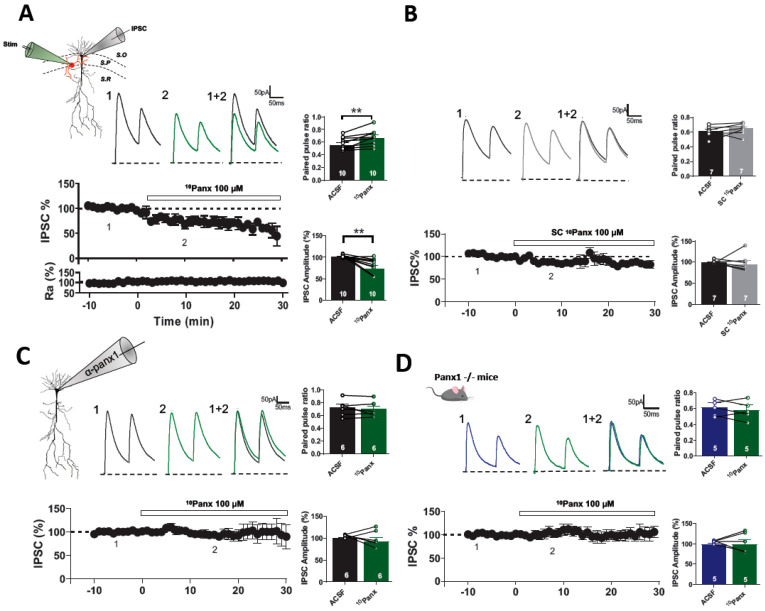
10panx depress evoked IPSC onto pyramidal neurons of mice hippocampus. (**A**) Schematic diagram showing the CA1 pyramidal neurons and localization of the stimulation and recording electrodes. In red, GABAergic interneuron and in black pyramidal neuron (**top**). Samples traces from evoked IPSCs and PPR for each condition (**top**). Temporal course showing the changes in the IPSC amplitude before, during and after 10panx (100 µM) application (**middle**). Temporal course showing Ra% along the experiment (**bottom**). Application of 10panx showed an increase of PPR at 100 ms intervals compared to the ACSF group. IPSC amplitude was significantly decrease with bath application of 10panx. (**B**) Samples traces from a representative experiment before and after sc10panx application (**top**). IPSC% vs time plot before and after sc10panx addition (**middle**). Temporal course showing Ra% along the experiment (**bottom**). (**C**) Representative traces of evoked IPSC by using α-panx1 (0.25 ng/µL) in the patch pipette before and after 10panx application (**top**). Temporal course showing IPSC% from cells recorded with α-panx1, before, during and after 10panx application (**bottom**). Bath application of 10panx had no effects in the PPR and IPSC amplitude in cells recorded with α-panx1 in the patch pipette. (**D**) Representative traces of evoked IPSCs for each condition in hippocampal slices from Panx1 KO mice (**top**). IPSC% vs time plot before and after 10panx addition in pyramidal neurons from Panx1 KO mice (**bottom**). Bath application of 10panx did not affect PPR and IPSC amplitude in hippocampal slices from Panx1 KO. Data are performed as mean ± SEM. ** *p* < 0.001. Number or recorded cells is indicated within bars.

**Figure 2 biomolecules-13-00887-f002:**
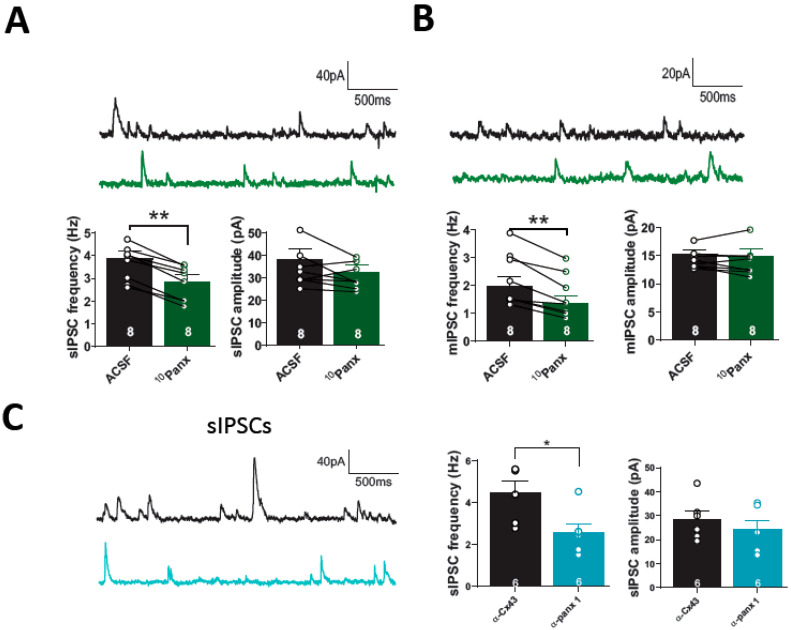
Block of Panx1 with 10panx decreases spontaneous GABAergic transmission onto pyramidal cells of mice hippocampus. (**A**) Samples traces of sIPSCs from pyramidal cells before (black) and after (green) 10panx application (**top**). Block of Panx1 decreased sIPSC frequency (**left**), while sIPSC amplitude was unchanged (**right**). (**B**) Representative sample traces showing mIPSCs recorded before and after bath application of 10panx (**top**). 10panx decreased mIPSC frequency compared with basal condition (**left**), while mIPSC amplitude remained unaffected (**right**). (**C**) Samples traces of sIPSCs obtained from pyramidal cells recorded with α-Cx43 (black) and α-panx1 (light blue) in the patch pipette (**left**). The frequency of sIPSCs was reduced in the presence of α-panx1 compared with its control condition, α-Cx43 (**middle**), while sIPSCs amplitude remained unaffected (**right**). Data are performed as mean ± SEM. * *p* < 0.05 and ** *p* < 0.01. Number or recorded cells is indicated within bars.

**Figure 3 biomolecules-13-00887-f003:**
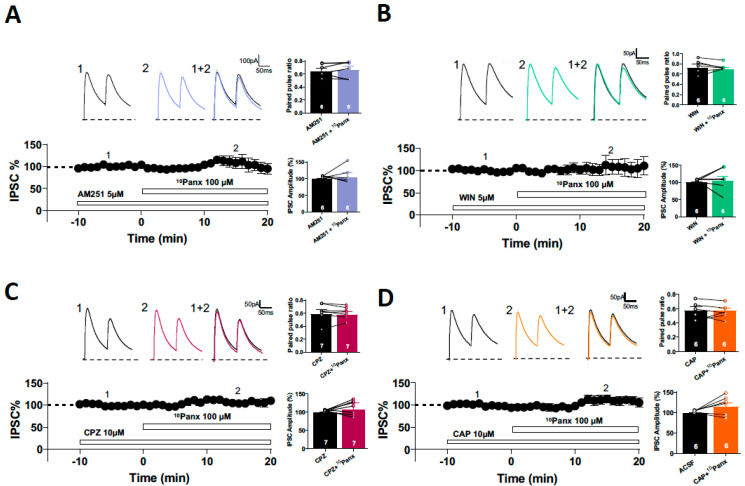
The blockage Panx1-dependent depression of inhibitory transmission requires the activation of CB1 receptors. (**A**) Samples traces from evoked IPSCs and PPR for each condition (**top**). Temporal course showing IPSC% before and after 10panx application, in the presence of AM251 (**bottom**). Bath application of 10panx had no effect neither in the PPR or in the IPSCs amplitude when the slice was pretreated with AM251. (**B**) Representative experiment for evoked IPSCs recorded for each condition (**top**). IPSC% vs time plot showing the changes in the IPSC% before, during and after 10panx application in the presence of WIN (**bottom**). Preapplication of WIN occluded the 10panx-mediated depression in the GABA release and IPSC amplitude. (**C**) Representative experiment for evoked IPSCs recorded for each condition (**top**). IPSC% vs time plot showing the changes in the IPSC% before, during and after 10panx application in the presence of CPZ (**bottom**). Preapplication of CPZ occluded the 10panx-mediated depression in the GABA transmission. (**D**) Representative experiment for evoked IPSCs (upper recordings) showing the IPSC% vs time plot showing the changes in the IPSC% before, during and after 10panx application in the presence of CAP. Preapplication of CAP blocked the 10panx-mediated depression in the GABA transmission. Data are performed as mean ± SEM. Number of recorded cells is indicated within bars.

**Figure 4 biomolecules-13-00887-f004:**
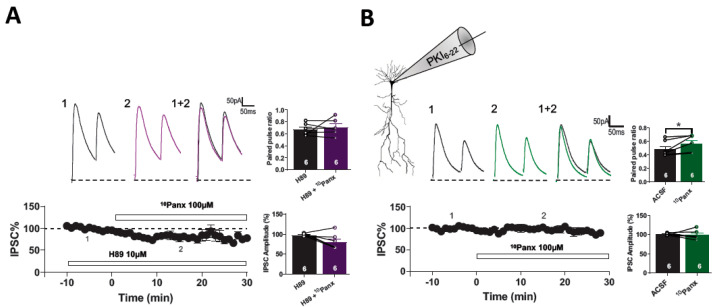
10panx-mediated depression in GABA release requires presynaptic cAMP/PKA signaling. (**A**) Samples traces from evoked IPSC and PPR for each condition (**top**). Temporal course showing the changes in the IPSC amplitude before, during and after 10panx (100 µM) application with the presence of PKA inhibitor H89 (10 µM) (**bottom**). H89 pre incubation blocked the10panx-induced increase in PPR and had no significant effect in IPSC amplitude in the presence of 10panx. (**B**) Representative traces from evoked IPSC and PPR for each condition with PKI_6-22_ peptide (2.5 µM) loaded in the recording pipette (**top**). On the bottom is showed IPSC% vs time plot before and after 10panx addition with PKI_6-22_ peptide. Loading PKI_6-22_ peptide failed to block the 10panx-mediated increase in paired pulse ratio. 10panx had no effect in the IPSC amplitude in cells recorded with PKI_6-22_ peptide in the recording pipette. Data are performed as mean ± SEM. * *p* < 0.05. Number of recorded cells is indicated within bars.

**Figure 5 biomolecules-13-00887-f005:**
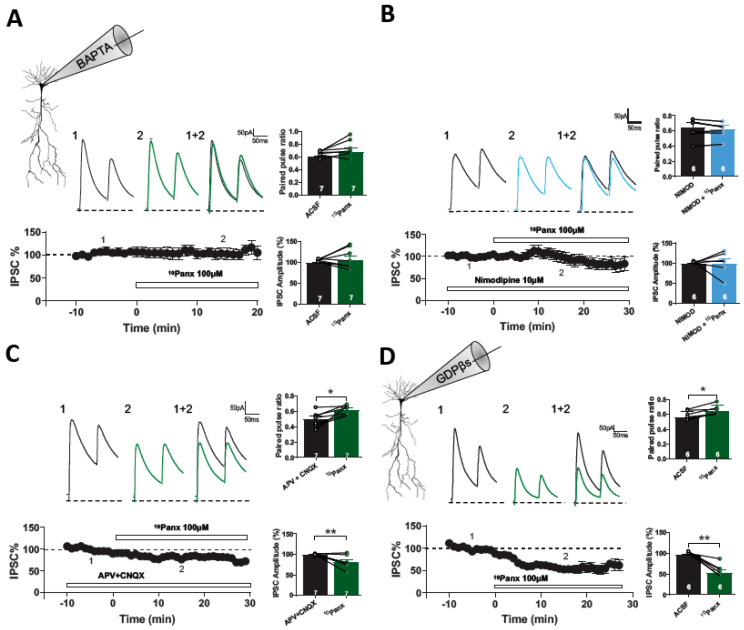
10panx-induced depression is independent of postsynaptic G-protein-coupled receptors but requires postsynaptic Ca^2+^ levels. (**A**) Samples traces from a representative experiment are shown on the **top**. Temporal course showing IPSC% before, during and after 10panx application in the presence of 20 mM BAPTA in the patch pipette (**bottom**). 10panx application increased PPR in cells recorded with 20 mM BAPTA in the recording pipette. 10panx-induced depression in the IPSC amplitude was blocked by 20mM BAPTA. (**B**) Representative experiment of evoked IPSCs from CA1 neurons pretreated with Nimodipine 10 µM (black) and after 10panx application (light blue) (**top**). IPSC% vs time plot showing the changes in the IPSC% before, during and after 10panx application in the presence of Nimodipine (**bottom**). Preapplication of Nimodipine occluded the 10panx-mediated depression in GABAergic efficacy because PPR and IPSC amplitude were unchanged. (**C**) Samples traces from evoked IPSC and PPR for each condition (**top**). Temporal course showing the changes in the IPSC amplitude before, during and after 10panx (100 µM) application in the presence of D-AP5 (20 µM) and CNQX (20 µM) (**bottom**). Application of 10panx showed a significant increase of PPR in the presence of APV and CNQX and it was accompanied of significant decrease in IPSC amplitude. (**D**) Samples traces from evoked IPSC and PPR for each condition (**top**). Temporal course showing IPSC% before, during and after 10panx with G-protein inhibitor GDP-βS (2 mM) in the recording pipette (**bottom**). With GDP-βS, bath application of 10panx significantly increased PPR and decreased IPSC amplitude. Data are performed as mean ± SEM. * *p* < 0.05 and ** *p* < 0.001. Number of recorded cells is indicated within bars.

**Figure 6 biomolecules-13-00887-f006:**
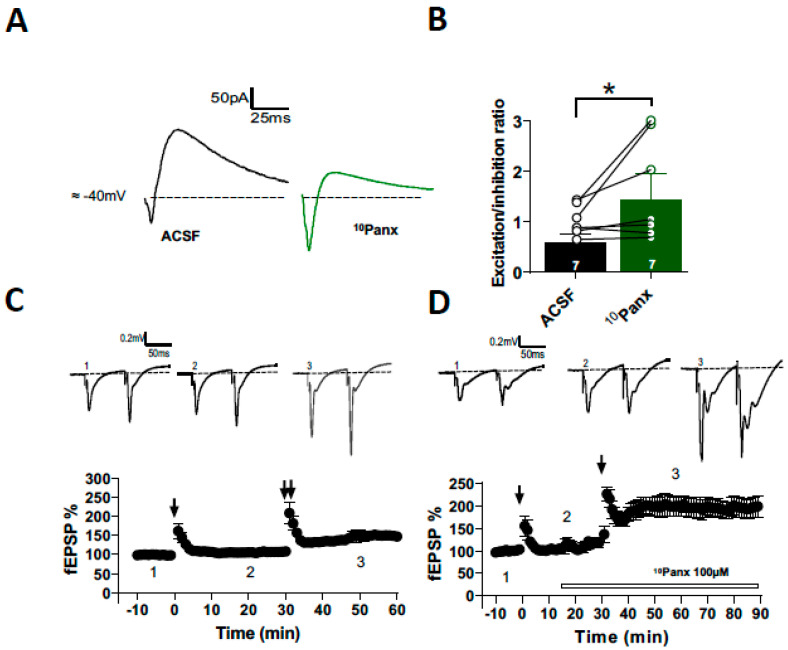
Block of Panx1 increases the excitation to inhibition ratio in CA1 pyramidal neurons of mice hippocampus. (**A**) Representative samples traces of recorded PSCs paired at −40 mV, before and after 10panx application. (**B**) E/I ratio calculated from peak amplitude either for GABAergic and glutamatergic component was increased in the presence of 100 µM 10panx. (**C**) Time course of fEPSP amplitude before (1), after single HFS (2) and 20 min after application of 2 HFS (3). Top panel shows representative averaged fEPSP recorded before (1), after single HFS (2) and 20 min after 2 HFS (3). (**D**) Time course of fEPSP amplitude before (1), after single HFS (2) and 20 min after single HFS under 10 Panx (3). Top panel shows representative averaged fEPSP recorded before (1), after single HFS (2) and 20 min after single HFS under 10 Panx (3). * *p* < 0.05.

## Data Availability

The data that support the findings of this study are available from the corresponding author upon reasonable request.

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
