# Peer review of "Pannexin-1 Modulates Inhibitory Transmission and Hippocampal Synaptic Plasticity"

_biomolecules, 2023, doi:10.3390/biom13060887_

Round 1

Reviewer 1 Report

The study by Garcia-Rojas aims to investigate the role of pannexin-1 (Panx1) in regulating GABAergic transmission and associated plasticity in the hippocampus. Overall, the study found that Panx1 reduces GABA transmission through a cannabinoid and calcium-dependent mechanism. These finding add to our understanding of synaptic physiology and mechanisms of plasticity. There are a few concerns that should be addressed:

1) Did AM251 (CB1R antagonist) change eIPSCs amplitude or PPR? The time course graph shows that 100% baseline is in the presence of AM251. However, if AM251 produced an effect, then the conclusion of an occluded effect can also be derived as it was for the WIN 55,212-2 experiment.

2) The observations that manipulating the postsynaptic neuron with either the anti-Panx1 antibody or the PKA-inhibitor, PKI6-22 affects PPR is interesting but somewhat contradictory to the effects of Panx1 being presynaptic. This would make more sense if Panx1-mediated increase in eCB release was dependent on PKA signaling in the postsynaptic neuron. This needs to be considered and discussed.

3) The conclusion that Panx1 has postsynaptic effects on GABA transmission is based on the observation that 10Panx1 reduced eIPSC amplitude. However, all of the other experiments, including PPR, sIPSC, and mIPSC show only changes in release mechanisms (i.e. PPR increases, sIPSC and mIPSC frequency decreases), while the typicaly classic postsynaptic measures, i.e. sIPSC and mIPSC amplitude, do not change. Based on this, it seems that the observed changes in eIPSC amplitude are likely a result of reduced GABA release.

Overall, the quality of English is good. However, all of the results are written in the present tense. It is recommended that the results section be revised and written in the past tense.

Author Response

Response to Reviewer 1 Comments.

The study by Garcia-Rojas aims to investigate the role of pannexin-1 (Panx1) in regulating GABAergic transmission and associated plasticity in the hippocampus. Overall, the study found that Panx1 reduces GABA transmission through a cannabinoid and calcium-dependent mechanism. These finding add to our understanding of synaptic physiology and mechanisms of plasticity. There are a few concerns that should be addressed:

Dear reviewer, we appreciate the feedback and valuable suggestions.

Point 1: Did AM251 (CB1R antagonist) change eIPSCs amplitude or PPR? The time course graph shows that 100% baseline is in the presence of AM251. However, if AM251 produced an effect, then the conclusion of an occluded effect can also be derived as it was for the WIN 55,212-2 experiment.

 Response 1: Thank you to reviewer. According to Supplementary Figure 2A (Page 20), it was observed that the administration of AM251 had no effect on the amplitude or paired-pulse ratio (PPR) of the synapses. This finding suggests that the release of endocannabinoids (eCB) at these synapses is dependent on demand. Notably, previous studies have indicated that AM251 inhibits the impact of eCB on the release of GABA or glutamate. As a result, the presence of AM251 in conjunction with 10panx1 did not have any significant effect on the amplitude of inhibitory postsynaptic currents (IPSCs) or the release of gamma-aminobutyric acid (GABA). We include in the line 240: “In the presence of the selective CB1 receptor antagonist AM251 (5 µM), which has been shown not to impact GABA transmission (supplementary Figure 2), the amplitude of evoked IPSCs (eIPSCs) was not significantly altered by the application of 10Panx1”.

Also, in supplementary Figure 2B (Page 20), it is illustrated that the administration of WIN results in a decrease in GABA transmission. Therefore, when WIN is perfused for a minimum of 10 minutes to ensure maximum impact on GABAergic transmission, the application of 10panx1 does not reduce GABA transmission. This indicates that the GABAergic depression induced by 10panx1 is completely blocked in the presence of WIN. We include in line 248 …(WIN) that depress the GABA transmission (supplemental figure 2B)...

Point 2: The observations that manipulating the postsynaptic neuron with either the anti-Panx1 antibody or the PKA-inhibitor, PKI6-22 affects PPR is interesting but somewhat contradictory to the effects of Panx1 being presynaptic. This would make more sense if Panx1-mediated increase in eCB release was dependent on PKA signaling in the postsynaptic neuron. This needs to be considered and discussed.

Response 2: We would like to express our gratitude for the question raised. In our study, PKI6-22 or α-panx1 was included in the recording pipette to block the postsynaptic PKA or postsynaptic Panx1, respectively. Between lines 442-451, we explore the possibility that PKA activity negatively regulates GABA receptor activity, depending on the specific subunit involved. It is important to note that the reviewer is correct in mentioning that PKA could also regulate the release of endocannabinoids (eCBs). We include in the discussion (line 429-432). Recent research has unveiled a noteworthy finding regarding the influence of postsynaptic PKA on the synthesis of 2-AG in striatal medium spiny neurons. This study further reveals that such activity has the potential to inhibit synaptic transmission in the cortico-striatal circuit [55]. However, investigating this specific relationship falls beyond the scope of our paper. Nonetheless, we acknowledge its significance and believe that further in-depth analysis of this association would be valuable for future research endeavors.

Point 3: The conclusion that Panx1 has postsynaptic effects on GABA transmission is based on the observation that 10Panx1 reduced eIPSC amplitude. However, all of the other experiments, including PPR, sIPSC, and mIPSC show only changes in release mechanisms (i.e. PPR increases, sIPSC and mIPSC frequency decreases), while the typically classic postsynaptic measures, i.e. sIPSC and mIPSC amplitude, do not change. Based on this, it seems that the observed changes in eIPSC amplitude are likely a result of reduced GABA release.

Response 3: We appreciate the reviewer's input. As mentioned in our previous response, between lines 442 -461, we extensively discuss the impact of both presynaptic and postsynaptic factors on GABA receptor function. We acknowledge that these effects may involve interactions with PKA or Panx1 activation, as well as potential modifications to GABA receptor function in the absence of these interactions.

Reviewer 2 Report

This is a very interesting article in which García-Rojas and colleagues demonstrate the involvement of Panexin 1 hemichannels in the efficacy of GABAergic transmission and therefore their importance in the hippocampus neuronal activity and synaptic plasticity.

The manuscript is well-written and illustrated, the experiments align with the proposed objectives, and the results are clear. The introduction provides sufficient information, and the results are thoroughly discussed. Taking all of this into account, from my perspective, this article meets the necessary standards for publication in its current form in the Biomolecules journal.

Author Response

Dear reviewer, we appreciate your positive comments.

Reviewer 3 Report

The study by García-Rojas et al. addresses the important issue of pannexin 1 role in inhibitory neurotransmission. The work convincingly shows that either the blockade or genetic deletion of pannexin 1 channels in CA1 pyramidal neurons through the activation of CB receptors leads to a decrease in GABAergic inhibition. Thus, the paper describes a new mechanism of pannexin 1 involvement in the control of excitation in the nervous system. The work was conducted at a high methodological level, the necessary controls were performed, which ensures the reliability of the results. In general, the work deserves publication after making minor changes, as well as some additions to the discussion.

1. The experiments were conducted at room temperature. It is known that the activity of ectonucletidases depends critically on temperature. Therefore, it should be discussed whether the mechanism proposed by the authors can be implemented at physiologically relevant temperature values.

2. The mechanism of induction of anandamide secretion by pannexin needs to be discussed in more detail.

Minor

Line 145-146: A high-frequency stimulation (HFS) protocol consisting of 145 1 train of 100 pulses at 100 Hz for 500 ms: should it be 1000 ms?

Line 27: blockage should be blockade. 

Author Response

Response to Reviewer 3 Comments

The study by García-Rojas et al. addresses the important issue of pannexin 1 role in inhibitory neurotransmission. The work convincingly shows that either the blockade or genetic deletion of pannexin 1 channels in CA1 pyramidal neurons through the activation of CB receptors leads to a decrease in GABAergic inhibition. Thus, the paper describes a new mechanism of pannexin 1 involvement in the control of excitation in the nervous system. The work was conducted at a high methodological level, the necessary controls were performed, which ensures the reliability of the results. In general, the work deserves publication after making minor changes, as well as some additions to the discussion:

Dear reviewer, we appreciate the feedback and valuable suggestions.

Point 1: The experiments were conducted at room temperature. It is known that the activity of ectonucletidases depends critically on temperature. Therefore, it should be discussed whether the mechanism proposed by the authors can be implemented at physiologically relevant temperature values.

Response 1: We would like to extend our gratitude to the reviewer for their valuable comment. Indeed, temperature is recognized as a crucial factor in regulating neuronal excitability and synaptic transmission. However, the specific effects of temperature on presynaptic release properties and post-synaptic receptors can vary across different studies. Therefore, we cannot discard that the magnitude and kinetics of the effects observed at room temperature may differ from the physiological temperature condition. In fact, it has been suggested that Panx1 channel activity increased with temperature (Palacios-Prado et al 2022). While our paper does not specifically address the temperature-dependent effects on Panx1, we recognize the importance of investigating the role of Panx1 under physiological temperature conditions in future studies. Examining the temperature sensitivity of Panx1 function would yield valuable insights into its regulatory mechanisms and its influence on GABAergic transmission. We are grateful for the suggestion provided by the reviewer and highlight its significance in advancing our understanding of Panx1-mediated processes in neuronal networks. As a result, we have included the following paragraph in our discussion section (line 461 highlighted in yellow). “Since we conducted the experiments at room temperature, it is possible that the magnitude and kinetics of the observed effects may differ under physiological temperature conditions. In fact, previous research [62] has shown that temperature increases the release of ATP from cells. Furthermore, it has been suggested that temperature also enhances the formation and function of intercellular channels formed by Panx1 in oligodendrocyte cell lines [3]. The authors observed a consistent temperature-dependent increase in Gap Junctional Conductance (gj) in these cells. Similarly, a previous study [63] demonstrated a temperature-dependent increase in hemichannel currents mediated by endogenous Cx38 in Xenopus oocytes. Therefore, we can speculate that the Panx1-mediated control of GABAergic transmission observed in acute experiments at room temperature may underestimate its actual magnitude under physiological temperatures”.

Point 2: The mechanism of induction of anandamide secretion by pannexin needs to be discussed in more detail.

Response 2: According to Bielecki et al. (2020) [11], Panx1 has been found to facilitate the clearance of anandamide (AEA) through its interaction with fatty acid amide hydrolase (FAAH). Consequently, the direct blockade of Panx1 results in increased tissue levels of AEA. Subsequently, AEA can act on presynaptic transient receptor potential vanilloid 1 (TRPV1) receptor, leading to enhanced calcium influx and facilitating activity-dependent glutamate release. These important findings, that were already in the previous version are now highlighted in yellow (line 61-64, 236-238, 254-256 and 434-437) in the revised version, provide support for the idea that Panx1 blockade contributes to a decrease in GABAergic synaptic efficacy by involving alterations at both the pre- and postsynaptic levels. It is important to acknowledge that our study did not specifically identify the endocannabinoid (eCB) involved in modulating GABA transmission. Further research is necessary to uncover the precise nature of the eCB that participates in this process.

Minor point: Line 145-146: A high-frequency stimulation (HFS) protocol consisting of 1 train of 100 pulses at 100 Hz for 500 ms: should it be 1000 ms?

Response to minor point: thank you, this mistake has been corrected in the new version (line 154; highlighted in yellow)
